# Cost-effectiveness analysis of implementing screening on preterm pre-eclampsia at first trimester of pregnancy in Germany and Switzerland

**Janne C. Mewes**[1], **Melanie Lindenberg**[1], **Hubertus J. M. Vrijhoef**[1,2]*

**1** Panaxea b.v., Amsterdam, North Holland, The Netherlands, **2** Maastricht University Medical Center, Limburg, The Netherlands

* bert.vrijhoef@panaxea.eu

## Abstract

### Objective

To assess the cost-effectiveness of preterm preeclampsia (PE) screening versus routine screening based on maternal characteristics in Germany and Switzerland.

### Methods

A health economic model was used to analyse the cost-effectiveness of PE screening versus routine screening based on maternal characteristics. The analysis was conducted from the healthcare perspective with a time horizon of one year from the start of pregnancy. The main outcome measures were incremental health care costs and incremental costs per PE case averted.

### Results

The incremental health care costs for PE screening versus routine screening per woman were €14 in Germany, and -CHF42 in Switzerland, the latter representing cost savings. In Germany, the incremental costs per PE case averted were €3,795. In Switzerland, PE screening was dominant. The most influential parameter in the one-way sensitivity analysis was the cost of PE screening (Germany) and the probability of preterm PE in routine screening (Switzerland). In Germany, at a willingness-to-pay for one PE case avoided of €4,200, PE screening had a probability of more than 50% of being cost-effective compared to routine screening. In Switzerland, at a willingness-to-pay of CHF0, PE screening had a 78% probability of being the most cost-effective screening strategy.

### Conclusion

For Switzerland, PE screening is expected to be cost saving in comparison to routine screening. For Germany, the additional health care costs per woman were expected to be €14. Future cost-effectiveness studies should be conducted with a longer time horizon.

**Data Availability Statement:** The authors confirm that the data supporting the findings of this study are available within the article and supporting

information files. Raw data were generated by
Panaxea.

**Funding:** Panaxea received an unrestricted grant
for this study from Thermo Fisher. The funders had
no role in the study design, data collection and
analysis, interpretation of data, decision to publish,
or preparation of the manuscript.

**Competing interests:** The authors have declared
that no competing interests exist. The expressed
views in this publication are solely the opinions of
the authors.

# Introduction

Preeclampsia (PE) is a hypertensive disorder, which manifests by high blood pressure and either proteinuria, maternal organ dysfunction, and/or uteroplacental dysfunction [1]. Preeclampsia can occur at term ($\geq$ 37 weeks of gestation) or preterm (before week 37 of gestation). Untreated, PE can progress to eclampsia; haemolysis, elevated liver enzymes, and low platelet count syndrome (HELLP-syndrome; or lead to placental abruption [1]. Most PE cases occur close to term and with mild symptoms [2]. PE developing with severe symptoms and during the preterm period is most problematic, as it is associated with more complications for the mother and the baby [2]. Especially as the only effective treatment of severe PE is the delivery of the baby, preterm PE is to be prevented. Worldwide, PE is a leading factor of morbidity and mortality of mothers and their babies [3]. PE can lead to an increased risk for foetal death [4], intrauterine growth restrictions, and preterm births with all its associated consequences [2]. In Europe, PE occurs in 0.4 to 2.8% of all pregnancies [2].

Several studies have shown that a prophylaxis with low-dose aspirin starting early in pregnancy lowers the incidence of PE. Guidelines of professional associations, such as the National Institute for Health and Clinical Excellence in the UK and the American College of Obstetricians and Gynaecologists, recommend aspiring prophylaxis with 100mg per day or more starting before or at week 16 of pregnancy for pregnant women at high risk of PE [5]. It is thus important to identify women at high risk of preterm PE.

In both Germany and Switzerland, antenatal care for pregnant women includes regular visits at the gynaecologist or, for pregnancies without complications, with a midwife. In the current standard of care, women at high risk of PE are identified through a routine screening during one of the first antenatal consultations, by means of analysing maternal characteristics and risk factors, as well as previous pregnancies and medical records. Known risk factors include a higher maternal age, PE during a previous pregnancy, obesity, nulliparity, or pre-existing hypertension [1]. Women identified as at high risk should be prescribed aspirin prophylaxis. However, by routine screening, most of the women at high risk of developing PE are not detected [5].

Additionally, an innovative screening for PE is available, conducted by a gynaecologist or specialist in the area of fetal medicine. Obstetrics history, maternal characteristics, mean arterial pressure (MAP), uterine artery pulsatility index (UAPI) evaluated in Doppler ultrasound, and the biomarkers serum placental growth factor (PlGF), and pregnancy-associated plasma protein-A (PAPP-A) measured at weeks 11–13 of gestation are used as an input parameter to the specific PE screening algorithm which allows to calculate individual risk for developing PE later on in pregnancy. Women identified as at high risk for PE should be prescribed aspirin prophylaxis [6]. For PE with delivery <37 weeks of gestation, a detection rate of PE screening of 80% was found with a false positive rate of 10% [6].

PE screening can thus well identify pregnant women who are at high risk of preterm PE, however, its implementation also adds costs to antenatal care. To inform implementation and reimbursement decisions of PE screening we conducted a cost-effectiveness analysis of PE screening in comparison to routine screening (the current standard of care) for Germany and Switzerland. The analysis was conducted from the health care perspective and adopted a time horizon of one year. This study focused on the prevention of preterm PE <37 weeks, as the performance of PE screening detecting women at high risk of PE $\geq$37 weeks is low [6].

## Materials and methods

A decision tree was constructed in Microsoft Excel to compare the cost-effectiveness of PE screening versus routine screening for all pregnant women in the first trimester in both Germany and Switzerland.

## Screening alternatives

In PE screening women were screened at 11+0 to 13+6 weeks of gestation, as recommended by the Fetal Medicine Foundation (FMF) [7]. Measurements taken were maternal characteristics, MAP, mean UAPI, the levels of PlGF and PAPP-A [1, 6]. An algorithm was used to predict the risk of preterm PE <37 weeks of gestational age (WGA) based on these measurements. Women with a risk of 1:100 or higher were classified as at high-risk [1]. All women at high-risk in PE screening were prescribed low-dose aspirin of 150mg/day starting before week 16 [5]. The Swiss guidelines state that the addition of PAPP-A to the screening would not lead to significant improvements in the screening and that most cases would also be detected without PAPP-A [8]. We therefore excluded it from the analysis for Switzerland, as it is unlikely it would be included in the screening in Switzerland when not recommended by the Swiss professional societies.

Routine screening consisted of analysing maternal characteristics during the anamnesis in one of the first antenatal appointments during the first trimester (usually between 8+0 to 10+6 weeks of gestation). Depending on the practice routine, patient characteristics, blood samples (protein and glucose), and blood pressure measurements were also used to identify a potential risk of developing PE during pregnancy [9, 10]. Women identified as being at high-risk for PE by their gynaecologist might be prescribed low-dose aspirin prophylaxis of 150mg/day starting before week 16 [5].

## Model structure

In the model, pregnant women either received PE screening or routine screening (Fig 1).

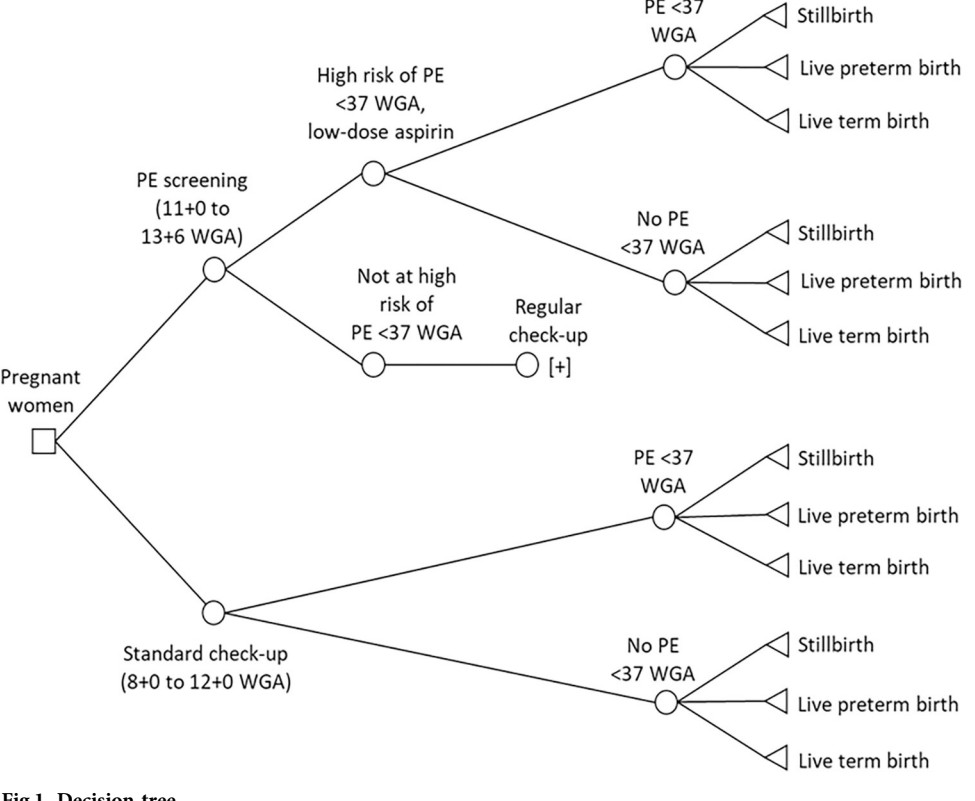

**Fig 1. Decision-tree.**

The standard check-up strategy includes routine screening on maternal characteristics

[+]: The branches from above are repeated

PE: preeclampsia, WGA: weeks of gestation.

In PE screening, women were classified as at high risk for preterm PE or not. Women at low risk of preterm PE received standard antenatal care. Women at high risk of preterm PE were prescribed low-dose aspirin prophylaxis. All women in PE screening could develop preterm PE or not, and have a stillbirth, a live preterm birth (<37 weeks of gestation), or a live term birth.

In routine screening, women could be classified as at high-risk based on maternal characteristics and might be prescribed low-dose aspirin prophylaxis. As no data was available on the probability of being classified as at high risk for preterm PE in routine screening and on subsequently receiving aspirin prophylaxis, we used summary data on the average number of women experiencing the outcomes stillbirth, live preterm birth, and live term birth. This was done irrespectively of a previous classification of being at high risk for preterm PE, as the available data did not allow for more level of detail.

The impact of false positive and false negative screening results for both screening alternatives were included in the outcomes of both interventions, i.e., some women received aspirin prophylaxis unnecessarily, and some did not receive aspirin prophylaxis whereas they should. This results in a higher chance in both screening alternatives of developing preterm PE than when the prophylaxis had been received.

The primary outcome measures were the incremental health care costs per women screened in PE screening versus routine screening and the incremental costs per PE case avoided. Secondary outcomes were the incremental costs per preterm birth avoided, and per still birth prevented. To assess the validity of the model structure it was presented separately to two (senior) physicians in the field of obstetrics and prenatal diagnostics (one in Germany, one in Switzerland).

## Input data

Input data were based on scientific literature, statistics databases, Diagnosis Related Group (DRG)-system databases, reports, and expert opinions. Country-specific data from Germany and Switzerland were used when available. For data that could not be identified in the literature, expert opinion was used (see above).

In PE screening, 11.03% of patients were classified as at high risk in both countries [5] and prescribed low-dose aspirin prophylaxis. Of the women at high-risk, 1.63% developed preterm PE in PE screening [5] and of those not at high risk, 0.18% [11].

In routine screening in Germany, 20% of pregnant women were classified as at high risk of preterm PE (expert opinion) of which 80% were assumed to be prescribed low-dose aspirin prophylaxis (expert opinion). In Switzerland, 10% were classified as high-risk in routine screening and 50% of these would receive low-dose aspirin prophylaxis (expert opinion). The risks of preterm PE in routine screening were based on Tan et al. (2018) [12], which represents a situation without using low dose aspirin prophylaxis. To account for receiving low-dose aspirin prophylaxis when being classified as high risk based on maternal characteristics in current practice, the incidence of preterm PE was reduced by 10.00% [13] to 0.72%.

In both screening alternatives, women could develop preterm PE or not, depending on the screening accuracy to detect preterm PE. Without preterm PE, the probabilities of caesarean section, stillbirth, and preterm birth were identical to the averages in the respective countries (see Table 1). When developing preterm PE, these were higher. For Germany, the probability of delivery by caesarean section with preterm PE was 99.2% [14], of stillbirth 0.74% [14], and

**Table 1. Input parameters: Risk of PE and treatment parameters.**

| Parameter | Value | Source |
|---|---|---|
| GERMANY | | |
| Yearly number of pregnancies | 763,732 | Calculation based on Federal Statistical Office 2020 [17, 18] |
| Routine screening | | |
| Probability being at high risk of PE | 20.00% | Expert opinion |
| Percentage of women at high risk receiving aspirin prophylaxis | 80.00% | Expert opinion |
| Probability of PE at <37 WGA | 0.72% | Tan 2018 [12], Askie 2007 [13] |
| Follow-ups at gynaecologist after giving birth | 1.00 | Assumption |
| Percentage of women attending follow-up after giving birth | 90.00% | Assumption |
| PE screening | | |
| Probability being at high risk of PE | 11.03% | Rolnik 2017 [5] |
| Percentage of women at high risk receiving aspirin prophylaxis | 100.00% | Rolnik 2017 [5] |
| Probability of PE at <37 WGA when at high risk | 1.63% | Rolnik 2017 [5] |
| Probability of PE at <37 WGA when at low risk | 0.18% | Rolnik 2017 [11] |
| Follow-ups at gynaecologist after giving birth | 1.00 | Assumption |
| Percentage of women attending follow-up after giving birth | 90.00% | Assumption |
| Probability of birth by caesarean section: | | |
| Women without preterm PE at <37 WGA | 29.10% | Federal Statistical Office 2020 [19] |
| Women with preterm PE at <37 weeks WGA | 99.20% | Bossung 2020 [14] |
| Probability of stillbirth: | | |
| Women without preterm PE at <37 WGA | 0.36% | Harmon 2015 [4] |
| Women with preterm PE at <37 weeks WGA | 0.74% | Bossung 2020 [14] |
| Probability of preterm birth: | | |
| Women without preterm PE at <37 WGA | 8.82% | Stubert 2014 [2] |
| Women with preterm PE at <37 weeks WGA | 75.00% | Bossung 2020 [14] |
| Additional length of stay mother with preterm PE | 3.0 | Ray 2017 [15] |
| Length of stay neonate after pregnancy with preterm PE | 16.0 | Ray 2017 [15] |
| SWITZERLAND | | |
| Yearly pregnancies | 84,759 | Calculation based on Pison 2015 [20], Federal Statistical Office 2021 [21, 22] |
| Routine screening | | |
| Probability being at high risk of PE | 10.00% | Expert opinion |
| Percentage receiving aspirin prophylaxis | 50.00% | Expert opinion |
| Probability of PE at <37 WGA | 0.72% | Tan 2018 [12], Askie 2007 [13] |
| Follow-ups at gynaecologist after birth | 1.00 | Assumption |
| Percentage of women attending follow-up | 100.00% | Assumption |
| PE screening | | |
| Probability being at high risk of PE | 11.03% | Rolnik 2017 [5] |
| Percentage receiving aspirin prophylaxis | 100.00% | Rolnik 2017 [5] |
| Probability of PE at <37 WGA when at high risk | 1.63% | Rolnik 2017 [5] |

*(Continued)*

**Table 1.** (Continued)

| Parameter | Value | Source |
|---|---|---|
| Probability of PE at <37 WGA when at low risk | 0.18% | Rolnik 2017 [11] |
| Follow-ups at gynaecologist after birth | 1.00 | Assumption |
| Percentage of women attending follow-up | 100.00% | Expert opinion |
| Probability of births by caesarean section: | | |
| Women without preterm PE at <37 WGA | 34.20% | Euro-Peristat Project 2018 [23] |
| Women with preterm PE at <37 weeks WGA | 61.88% | Calculation based on Hodel 2020 [16] |
| Probability of stillbirth: | | |
| Women without preterm PE at <37 WGA | 0.36% | Harmon 2015 [4] |
| Women with preterm PE at <37 weeks WGA | 0.74% | Harmon 2015 [4] |
| Probability of preterm birth: | | |
| Women without preterm PE at <37 WGA | 6.70% | Purde 2015 [3] |
| Women with preterm PE at <37 weeks WGA | 75.00% | Stubert 2015 [2] |

PE: preeclampsia, WGA: weeks of gestation

of preterm birth 75.0% [14]. With preterm PE, the length of stay increased by 3 days for the mother and 16 days for the neonate [15]. For Switzerland, the probability of delivery by caesarean section of women with preterm PE was 61.9% (calculation based on [16]), of stillbirth 0.74% [4], and of preterm birth 75.00 [3]. For Switzerland, the extended length of stay for preterm PE patients was included using a DRG-code for a hospital stay with preterm PE. All input data are shown in Table 1.

Costs incurred by patients included the costs of the regular antenatal care consultation; of PE screening; of the aspirin prophylaxis; of treatment including admission to the hospital for the mother when having preterm PE; of giving birth and the hospital stay of the mother and the baby after giving birth; and, if applicable, the costs of stillbirth, admission of a preterm born neonate, additional health care costs during the first months of life of a preterm neonate; and follow-up costs including a regular check-up visit at the gynaecologist of the mother a couple of weeks after giving birth.

The specific costs of PE screening were not known as in both countries it currently is offered privately, with prices likely being higher as when included in public health insurance. The costs were thus based on current reimbursement for the biomarkers combined with estimates of the experts. For Germany, screening costs of €90 were used and for Switzerland costs of CHF150. All cost input parameters were inflated to their 2021 value and are shown in Table 2.

## Data-analysis

Per screening method, the probability of patients following each branch along the decision-tree was calculated. For analysing costs, the probabilities of following each branch were multiplied with the costs incurred along the way. Per screening method, the health care costs, the number of PE cases, and the number of stillbirths were summed up. The results were presented as averages per woman screened and on the population level for all pregnant women per year in the respective country, which were 763,732 in Germany [17] and 84,759 in Switzerland [20, 21].

For the incremental difference, the outcomes of the routine screening group were subtracted from those of the PE screening group. To arrive at the incremental costs per PE case averted of PE screening versus routine screening, the incremental costs were divided by the incremental number of PE cases.

**Table 2. Cost input parameters.**

| Parameter | Unit cost | Source |
|---|---|---|
| GERMANY | [EUR] | |
| PE screening | 90.00 | Consists of €19.40 for PlGF (ebm 32362), and for PAPP-A, and an assumed €62.35 for drawing the blood sample, Doppler ultrasound, MAP, and counselling. |
| Regular pregnancy check-ups, per quartile | 130.00 | KBV code 01770 |
| Low-dose aspirin prophylaxis | 11.37 | 150mg/day for 22 weeks. Average price of multiple manufacturers and sellers |
| Stay at hospital in case of PE, per day | 1,170.99 | Pokras 2018 [24] |
| Caesarean section including hospital stay | 3,443.93 | Average of DRG codes O01D, O01E, O01F, and O01G |
| Vaginal birth including hospital stay | 2,100.99 | Average of DRG codes O60B, O60C, and O60D |
| Hospital stay of healthy new-born | 703.68 | Average of DRG codes P67E and P66D |
| Stillbirth | 1,515.83 | Mistry 2013 [25]. Includes bereavement counselling, autopsy, and placental pathology |
| Stay at NICU, per day | 1,536.23 | Martin 2008 [26]. NICU costs assumed to be similar to adult ICU costs. |
| Hospital stay neonate born <37 WGA | 22,257.22 | LOS from Ray 2017 [15] (16 days) corrected for a normal term birth (2 days) multiplied with the NICU costs. |
| Additional healthcare in first 3 months of life of preterm born | 1,675.49 | Calculation based on Jacob 2017 [27] |
| Visit to outpatient clinic after delivery | 130.00 | Assumed. Per quartile. |
| SWITZERLAND | [CHF] | |
| PE screening | 150.00 | Assumption, including PlGF (Sfr. 80, tariff code 1474.10) and Doppler ultrasound. |
| Regular pregnancy check-up at gynaecologist | 35.72 | TARDOC 1.1. HF0002. 7 visits during pregnancy assumed. |
| Ultrasound during pregnancy check-up | 141.86 | TARDOC 1.1 PW1001. 2 ultrasounds assumed. |
| Low-dose aspirin prophylaxis | 22.89 | 150mg/day for 22 weeks. Average price of multiple manufacturers and sellers |
| Stay at hospital in case of PE, per stay | 3,986.42 | DRG O65C |
| Caesarean section including hospital stay | 9,268.93 | Average of DRGs O01E and O01G |
| Vaginal birth | 6,158.29 | Average of DRGs O60A, O60D |
| Admission healthy new-born | 2,133.04 | DRG P67D |
| Stillbirth | 2,417.36 | Mistry 2013 [25]. Includes bereavement counselling, autopsy, and placental pathology. Adjusted by purchasing power parity from United Kingdom to Switzerland. |
| Hospital stay neonate born at <37 WGA | 67,837.32 | Weighted averages of DRGs: P05A, P05B, P03A, P03B, P63Z, P04B, P04C, and P65A |
| Additional healthcare in first 3 months of life of preterm born | | Calculation based on Jacob 2017 [27] |
| Visit to outpatient clinic after delivery | 49.13 | TARDOC 1.1. HF0007 |

DRG: diagnosis-related group, ICU: intensive care unit, KBV: German National Association of Statutory Health Insurance Physicians, NICU: neonatal intensive care unit

## Sensitivity and scenario analyses

To analyse the robustness of the results, a one-way sensitivity analysis was conducted in which each input parameter was varied individually by its standard deviation, +/- 25%, or a

reasonable range (see S1 Table). The top ten parameters influencing the incremental costs the most were shown in a tornado diagram.

To analyse the uncertainty surrounding the results, a probabilistic sensitivity analysis (PSA) was conducted on the incremental costs per PE case avoided. This outcome measure was chosen since the main aim of the screening is to prevent cases of preterm PE, and as developing preterm PE also includes the higher chance of developing preterm birth or still birth, as well as the negative health consequences of preterm PE itself. Healthcare costs and the number of PE cases in PE screening and routine screening were drawn from distributions reflecting the likely range of parameters in 5,000 iterations (Monte Carlo Simulation). For probabilities we used beta distributions and for costs and resource use gamma distributions. The results were shown on a cost-effectiveness plane and in a cost-effectiveness acceptability curve (CEAC). For the CEACs, the proportion of drawn values that remains below certain willingness-to-pay-thresholds was calculated [28].

In a scenario analysis we evaluated the effect of conducting a variant of the contingent PE screening in which UAPI (measured by Doppler) is only offered to the 30% of patients at highest risk. This reduces the PE screening costs to €58 for Germany and to CHF101 for Switzerland. The detection rate when conducting PE screening without the UAPI is expected to be similar but slightly lower and depends on the population screened [29]. However, a study on the actual number of preterm PE cases prevented with the contingent screening is not available. To conservatively account for a potentially lower detection rate, we increased the rate of PE cases at the same time by 10% as compared to when offering PE screening with all components to not underestimate the impact of contingent screening.

### Ethics statement

As this study does not involve patients or study subjects, according to the Dutch Medical Research in Human Subject Act (WMO), it is exempt from ethical approval in the Netherlands.

## Results

### Base case results Germany

The average cost per women were €5,916 in PE screening and €5,901 in routine screening, leading to incremental costs for PE screening of €14 per pregnant women (difference due to rounding). The percentage of women developing preterm PE decreased from 0.71% in routine screening to 0.34% in PE screening, leading to an ICER of 3,795€ per PE case averted. For 763,732 women pregnant in Germany per year, the number of stillbirths decreased by 11 in the PE screening group compared to routine screening, and the number of PE cases by 2,891. See Table 3 for all results.

### Base case results Switzerland

For Switzerland, the average healthcare cost per woman were CHF14,800 in PE screening and CHF14,842 in routine screening, leading to cost-savings per woman screened on preterm PE of CHF42. The percentage of preterm PE cases in routine screening was 0.72% in routine screening and 0.34% in PE screening. For all 84,759 pregnant women in Switzerland in one year, one still birth could be prevented and 321 preterm PE cases, for PE screening versus routine screening. As PE screening in Switzerland was expected to be cost saving and leading to fewer cases of preterm PE, stillbirth, and preterm birth, it dominated the standard of care for

**Table 3. Results for Germany and Switzerland.**

| Germany: Results per patient | | | | |
|---|---|---|---|---|
| | **Costs [EUR]** | **Preterm PE cases** | **Preterm birth** | **Stillbirths** |
| PE screening | 5,916 | 0.0034 | 0.0905 | 0.00361 |
| Routine screening | 5,901 | 0.0072 | 0.0930 | 0.00363 |
| Incremental (PE screening vs. RS) | 14 | -0.0038 | -0.0025 | -0.000015[a] |
| ICER | N/A | €3,795/ PE case averted | €5,734/ preterm birth averted | €990,316/ stillbirth averted |
| **Germany:** Results for all pregnant women per year | | | | |
| | **Costs [EUR]** | **Preterm PE cases** | **Preterm birth** | **Stillbirths** |
| PE screening | 4,518,010,541 | 2,608 | 69,087 | 2,759 |
| Routine screening | 4,507,040,577 | 5,499 | 71,000 | 2,771 |
| Incremental (PE screening vs. RS) | 10,969,963 | -2,891 | -1,913 | -11 |
| ICER | N/A | €3,795/ PE case averted | €5,734/ preterm birth averted | €990,316/ stillbirth averted |
| **Switzerland:** Results per patient | | | | |
| | **Costs [CHF]** | **Preterm PE cases** | **Preterm birth** | **Stillbirths** |
| PE screening | 14,800 | 0.0034 | 0.0693 | 0.00361 |
| Routine screening | 14,842 | 0.0072 | 0.0719 | 0.00363 |
| Incremental (PE screening vs. RS) | -42 | -0.0038 | -0.0026 | -0.000015[a] |
| ICER | N/A | Dominant | Dominant | Dominant |
| **Switzerland:** For all pregnant women per year | | | | |
| | **Costs [CHF]** | **Preterm PE cases** | **Preterm birth** | **Stillbirths** |
| PE screening | 1,254,424,491 | 289 | 5,877 | 306 |
| Routine screening | 1,257,955,337 | 610 | 6,096 | 307 |
| Incremental (PE screening vs. RS) | -3,530,846 | -321 | -219 | -1 |
| ICER | N/A | Dominant | Dominant | Dominant |

ICER: Incremental cost-effectiveness ration, PE: preeclampsia, RS: Routine screening

[a] Numbers may not add up due to rounding.

incremental costs per preterm PE case, stillbirth, and preterm birth avoided. See Table 3 for all results.

## Sensitivity analysis

In the one-way sensitivity analysis, for Germany, the most influential parameters were (1) the cost of PE screening, (2) the probability of preterm PE in standard of care, and (3) the probability of preterm birth with preterm PE. For Switzerland, the incremental costs were most influenced by varying (1) the probability of preterm PE in routine screening, (2) the probability of preterm birth with preterm PE, and (3) the cost of a hospital stay for a neonate born prematurely. The tornado diagrams (Figs 2 and 3) show the ten most influential parameters per country.

The results of the PSA were plotted on the cost-effectiveness planes. In Germany, most values were in the top left quadrant, indicating fewer PE cases against higher costs. The remaining values were plotted in the bottom left quadrant, indicating fewer PE cases against lower costs than with routine screening. In Switzerland, most values were in the bottom left quadrant, indicating fewer PE cases and lower healthcare costs in PE screening versus routine screening. The remaining values drawn were in the upper left quadrant, indicating fewer PE cases against higher costs for PE versus routine screening, see Figs 4 and 5.

The CEACs show that in Germany, PE screening had a higher probability to be cost-effective in comparison to routine screening at a willingness-to-pay rate of €4,200 per PE case

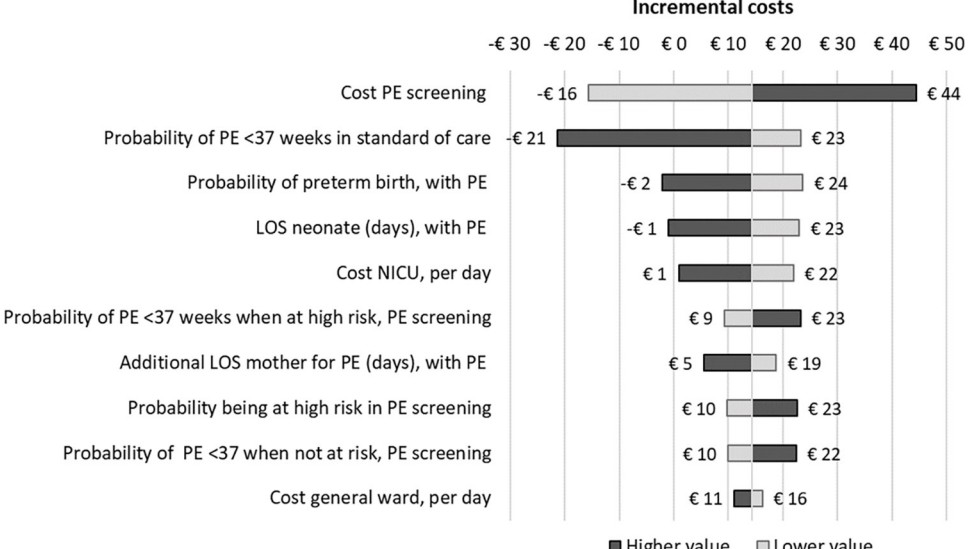

**Fig 2. Tornado diagram Germany.** Base case value is €14. PE relates to preterm PE <37 weeks of gestation.

avoided (Fig 6). In Switzerland, PE screening had a 78% probability of being the most cost-effective screening strategy at a willingness-to-pay of 0 CHF for one PE case avoided (Fig 7).

The scenario analysis showed that introducing contingent PE screening with offering the UAPI measurement to only the 30% of patients being at highest risk would result in incremental costs per woman screened for PE screening versus routine screening of -€11 in Germany and -CHF73 in Switzerland, indicating cost-savings for both countries.

## Discussion

In this study the cost-effectiveness of PE screening was analysed for all pregnant women in the first trimester compared to the routine screening based on maternal characteristics, for

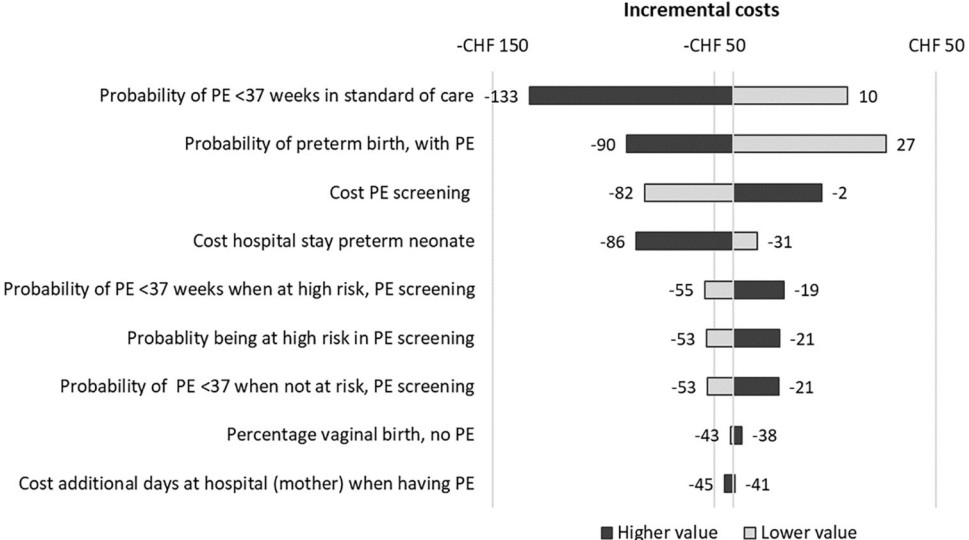

**Fig 3. Tornado diagram Switzerland.** Base case value is: -CHF 42. PE relates to preterm PE <37 weeks of gestation.

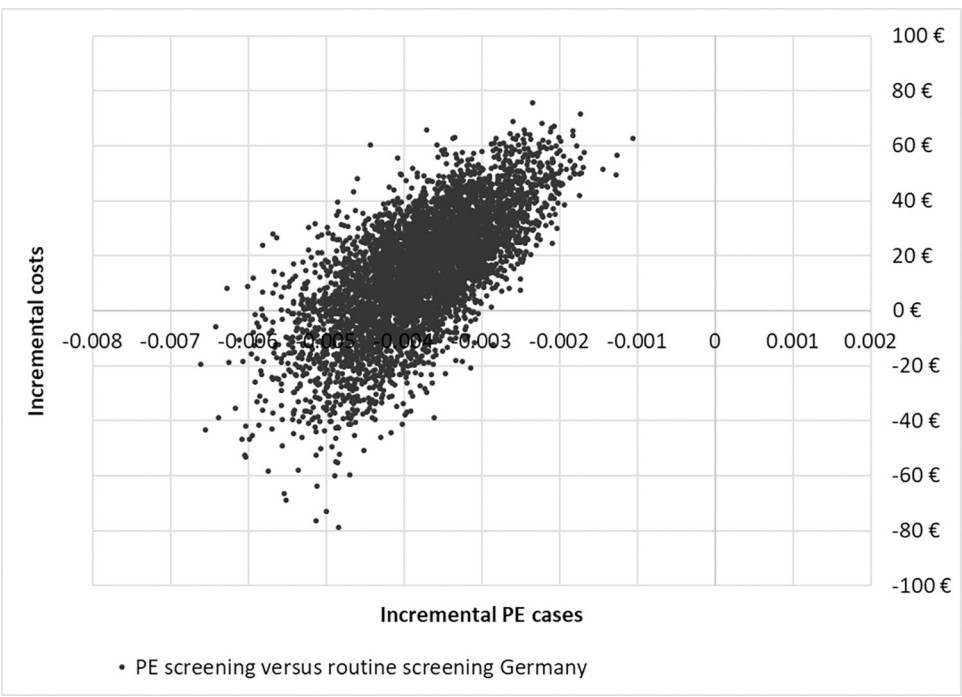

**Fig 4. Cost-effectiveness plane Germany.**

Germany and Switzerland. In both countries, PE screening reduced the number of PE cases and the number of stillbirths. The average incremental healthcare costs per women undergoing PE screening versus routine screening was €14 in Germany and -CHF42 in Switzerland,

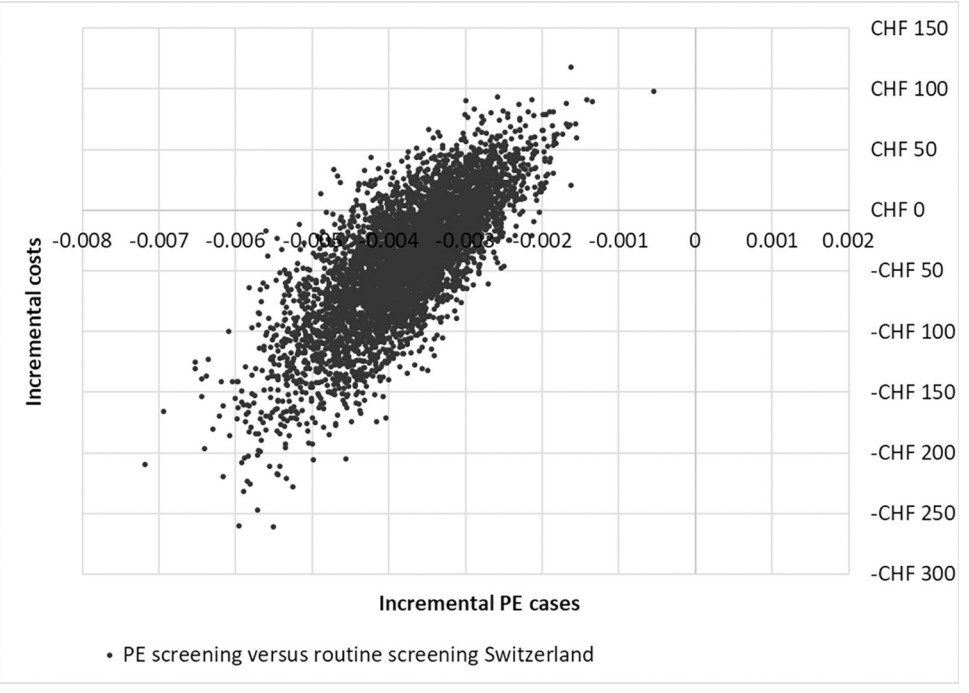

**Fig 5. Cost-effectiveness plane Switzerland.**

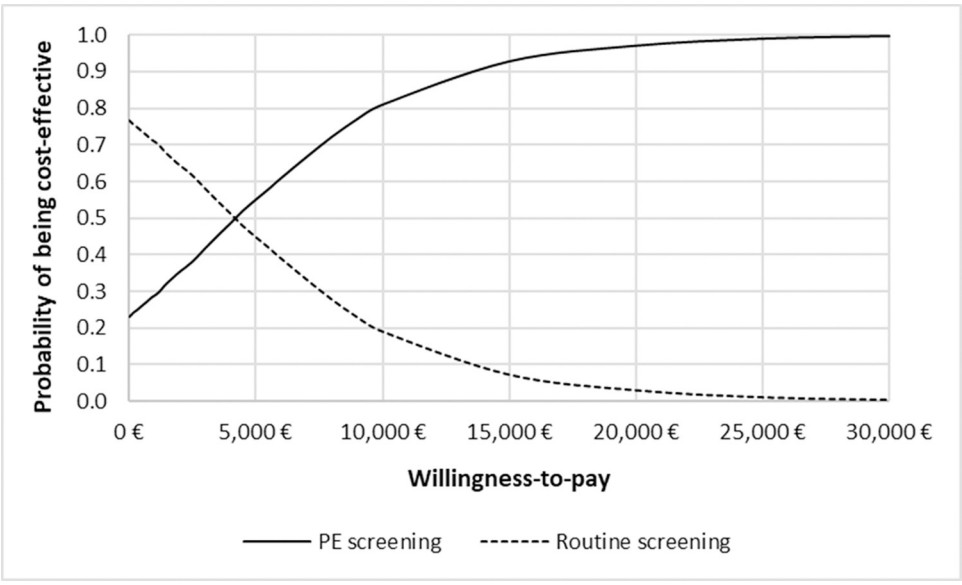

**Fig 6. Cost-effectiveness acceptability curve for Germany.**

the latter representing cost savings. The ICER of costs/PE case averted was €3,795 in Germany for PE screening versus routine screening, while PE screening dominated in Switzerland. The incremental costs of PE screening in Germany were mainly driven by the probability of developing preterm PE in routine screening, the screening costs, and the probability of preterm birth when having preterm PE. For Switzerland, the drivers of the incremental costs were the probability of developing preterm PE in routine screening, followed by the probability of preterm birth with preterm PE and the costs of the hospital stay of neonates born preterm.

Several earlier studies have evaluated the economic benefit of PE screening from a non-European perspective. A study from the US societal perspective compared no aspirin use to PE

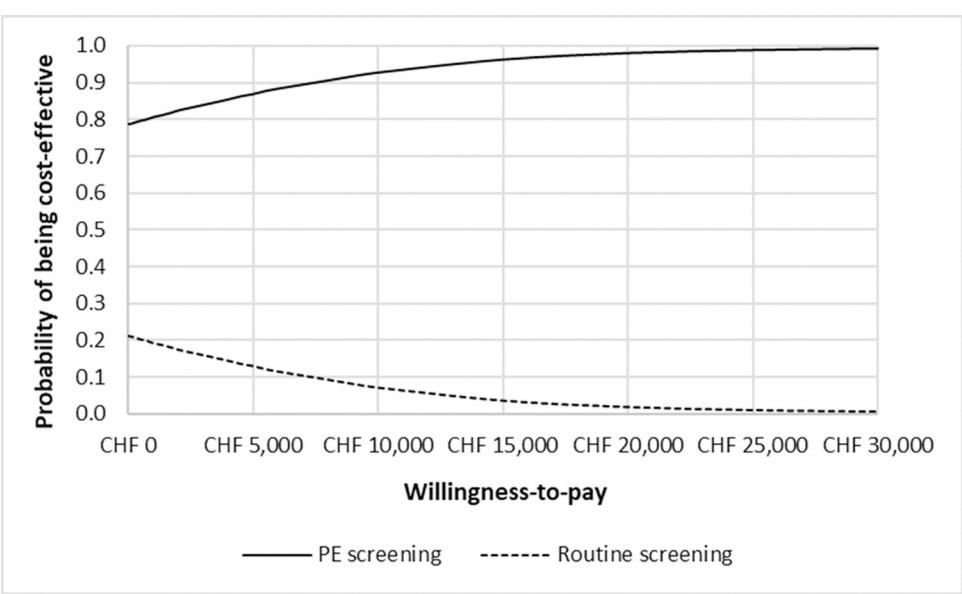

**Fig 7. Cost-effectiveness acceptability curve for Switzerland.**

screening and prophylactic aspirin, to screening based on maternal characteristics and to universal aspirin use [30] Universal aspirin use was dominant with fewer costs and fewer cases of PE compared to the other strategies. PE screening had higher total costs and fewer PE cases than screening based on maternal characteristics [30]. A Canadian study, conducted from the healthcare perspective over a one-year time horizon compared PE screening and prophylactic aspirin for those at high risk to current practice [31]. PE screening was found to be cost saving, preventing 1,096 early-onset PE cases/year at a cost saving of C$14.3m [31]. These previous publications both reported a reduction in the number of PE cases, with the cost-effectiveness depending on the costs included in the analysis and the willingness-to-pay for the prevention of PE screening.

The current study has a couple of limitations. First, the input data for the percentage of women who develop preterm PE in standard of care was relatively old [12]. It is possible that with time gynaecologists in routine screening would prescribe aspirin prophylaxis to an increasing percentage of women at high risk of preterm PE. Therefore, in the data for routine screening, fewer women identified as at high risk received aspirin prophylaxis than they would in current practice. We therefore adjusted the percentage by a relative 10% [13]. Second, the costs of PE screening are based on an estimation,. Depending on the amount of reimbursement, the PE screening costs might deviate, and influence the incremental costs per woman screened. Third, the time horizon of this economic evaluation of one year was not able to capture the negative long-term consequences of PE for the mother and the baby. Though the data still are limited, these include a higher risk of the mother for kidney disease [32], hypertension, ischemic heart disease, thromboembolism, and diabetes type 2 [33, 34]. The risks associated with preterm birth of the neonate include bronchopulmonary dysplasia, cerebral palsy, and an increased long-term risk of diabetes type 2, obesity, and cardiovascular disease [1]. Considering the costs of managing these long-term consequences by conducting an economic analysis with a longer time horizon could increase the chance of PE screening being cost-effective, assuming the relationship between PE and the long-term consequences is causative. Based on the results of this analysis, observing a reduction in the number of preterm PE cases, it is recommended to implement PE screening in practice and to include it in the reimbursed care. The additional costs of €14 per pregnant woman in Germany are relatively low considering the potential to prevent preterm PE cases, stillbirths, and especially considering the long-term health consequences of PE for the mother and for the babies born preterm. Considering equal access to care it is desirable that all women have the possibility to participate in the screening, regardless of their possibility to pay for it out of pocket, which currently is the case. In addition, access would be improved when all gynaecologists would offer PE screening which requires special education and training. Now, patients mostly are referred to specialists.

Some of the measurements required for PE screening, the ultrasound markers, and the biomarkers PAPP-A and PlGF (as optional marker), are also conducted for the first trimester trisomy screening. In Switzerland, the first trimester screening on trisomy is reimbursed for all pregnant women and approximately 90% of woman opt to conduct it (expert opinion). It is thus possible to use the measurements conducted for trisomy screening also for PE screening, which resulted in lower screening costs compared to Germany.

For Switzerland, PE screening was found to be cost saving whereas for Germany low additional costs per woman screened were found. The reason for this difference lies in the costs used in the analysis, which differ between Germany and Switzerland. The estimated cost of PE screening differs, as well as the costs of hospitalisation for a preterm born neonate. In Switzerland, the cost input used for hospitalisation of a preterm born neonate were higher than the input used for Germany. For Germany, these costs were based on the average length of stay

and the NICU costs per day. As no specific cost for a day on the NICU could be identified, that of a standard ICU-day was used, which might underestimate the actual cost.

Currently, a discussion is ongoing on the parameters the PE screening should consist of. Including all parameters (i.e., maternal factors, MAP, UAPI, PAPP-A, and PlGF) leads to the highest detection rate for preterm preeclampsia <37 weeks of gestation. Removing one or two parameters from the screening results in a small decrease in the detection rate [6]. However, leaving out the more costly parameter, such as UAPI, would significantly decrease the PE screening costs, while influencing the detection rate only be small percentages [6], resulting in a higher chance for PE screening to be judged as cost-effective.

Earlier economic evaluations showed that the most cost-effective strategy is to prescribe low-dose aspirin to all pregnant women without conducting a specific screening on PE [30]. It is not expected that low-dose aspirin during pregnancy leads to considerable side effects, adverse events, or damage to the baby. However, the economic analysis which included this strategy also accounted for a higher risk of gastrointestinal bleeding and aspirin-exacerbated respiratory disease [30]. Another disadvantage of the universal strategy might be a decrease in compliance to aspirin intake. Pregnant women who do not know to be at higher risk for PE might not be willing to take aspirin, especially as pregnant women are generally (asked to be) very cautious with medication. The question is if those being compliant finally correspond with the group at highest risk.

In the prevention and management of preeclampsia after the first trimester, screening alternatives are available next to routine and innovative PE screening in the first trimester. For short-term prediction of preeclampsia, novel maternal serum biomarkers, such as the ratio between anti-angiogenic soluble fms-like tyrosine kinase-1 (sFlt-1) and PlGF taken between 18 +0 and 36+6 weeks of gestation were found suitable to identify pregnant women with PE symptoms who require hospitalisation and women who can safely be monitored as outpatients [35]. A review by Schlembach et al. (2019) identified that angiogenic biomarkers to predict preeclampsia are cost-saving, among others by preventing unnecessary hospitalisation [36]. Thus, while PE screening is suitable for identifying women at high risk of preterm PE in the first trimester who benefit from aspirin prophylaxis, angiogenic biomarkers form cost-saving options for predicting the short-term risk of PE between 18+0 to 36+6 weeks of gestation.

In conclusion, this study showed that PE screening was expected to reduce the number of preterm PE cases and of stillbirths. For Switzerland, PE screening was found to be cost saving compared to the standard of care. In Germany, additional costs of €14 per woman screened were expected. In the interpretation of these results, it should be considered that the prevention of long-term health consequences caused by preterm births and potentially by preterm PE were not included within the one-year time horizon of the analysis. Further research focused on the occurrence of long-term health effects of PE for women and their children is required to be able to include potential long-term cost savings by preventing preterm PE into economic analyses of PE screening in the first trimester of pregnancy.

## Supporting information

**S1 Table.**
(DOCX)

## Acknowledgments

The authors like to thank the two experts in fetal medicine physicians from Germany and Switzerland for providing their expert opinions.

## Author Contributions

**Conceptualization:** Hubertus J. M. Vrijhoef.

**Formal analysis:** Janne C. Mewes, Melanie Lindenberg.

**Funding acquisition:** Hubertus J. M. Vrijhoef.

**Investigation:** Janne C. Mewes.

**Methodology:** Janne C. Mewes, Hubertus J. M. Vrijhoef.

**Project administration:** Janne C. Mewes, Hubertus J. M. Vrijhoef.

**Supervision:** Hubertus J. M. Vrijhoef.

**Writing – original draft:** Janne C. Mewes, Melanie Lindenberg.

**Writing – review & editing:** Hubertus J. M. Vrijhoef.

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
