## [Decision Letter · Decision Letter 0]

18 Nov 2021

PONE-D-21-23946Cost-effectiveness analysis of implementing screening on preterm pre-eclampsia at first trimester of pregnancy in Germany and SwitzerlandPLOS ONE

Dear Dr. Vrijhoef,

Thank you for submitting your manuscript to PLOS ONE. After careful consideration, we feel that it has merit but does not fully meet PLOS ONE’s publication criteria as it currently stands. Therefore, we invite you to submit a revised version of the manuscript that addresses the points raised during the review process.

We look forward to receiving your revised manuscript.

Kind regards,

Rashidul Alam Mahumud, MPH, MSc, PhD

Academic Editor

PLOS ONE

4. PLOS requires an ORCID iD for the corresponding author in Editorial Manager on papers submitted after December 6th, 2016. Please ensure that you have an ORCID iD and that it is validated in Editorial Manager. To do this, go to ‘Update my Information’ (in the upper left-hand corner of the main menu), and click on the Fetch/Validate link next to the ORCID field. This will take you to the ORCID site and allow you to create a new iD or authenticate a pre-existing iD in Editorial Manager. Please see the following video for instructions on linking an ORCID iD to your Editorial Manager account: https://www.youtube.com/watch?v=_xcclfuvtxQ.

Additional Editor Comments:

Please revise your manuscript according to the reviewer's advise and suggestions. Table 3 and 4 may be combined with one Table.

Reviewers' comments:

Reviewer's Responses to Questions

**Comments to the Author**

1. Is the manuscript technically sound, and do the data support the conclusions?

Reviewer #1: Yes

2. Has the statistical analysis been performed appropriately and rigorously? 

Reviewer #1: Yes

3. Have the authors made all data underlying the findings in their manuscript fully available?

Reviewer #1: Yes

4. Is the manuscript presented in an intelligible fashion and written in standard English?

Reviewer #1: Yes

5. Review Comments to the Author

Reviewer #1: This article describes a model-based cost-effectiveness analysis of screening for pre-term eclampsia compared to usual care in Germany and Switzerland. The question addressed is interesting, as the screening procedure is currently not covered by statutory health insurers in Germany and Switzerland and published economic evaluation studies for these countries are lacking.

Overall, the article is well-structured. However, the following points require further specification:

-Some background on the different health care and policy contexts in Germany and Switzerland (e.g., what does usual care include, who conducts screening etc.) would be helpful in the introduction. In line 95 an explanation should be added why PAPP-A was not recommended for Switzerland.

-The decision tree does not distinguish all possible scenarios. For example, false positive and false negative screening tests and aspirin vs. no aspirin among patients with a standard check-up are not included. Possibly, only summary data are available which do not allow more level of detail in the model. Please provide a rationale for the model structure.

-The role of the variable “pre-term birth” is unclear. In the decision tree it seems that pre-term birth is an outcome, however it is not included in table 3 and 4. Pre-term births and still births averted seem a more meaningful outcome from the patient perspective than PE cases averted. Please describe the rationale for choosing PE cases averted as primary outcome.

-In Figure 2 a legend would be helpful. I assume the (+) means that the path starting with standard check up needs to be added here.

-Information is missing about key assumptions regarding the relationship between events and costs. For example, one would assume that the rate of Caesarian sections is higher among patients with PE.

-The term “interventions” is confusing. It would be clearer to speak about alternatives or more specifically about routine screening and innovative screening. According to my understanding screening does already take place in routine care, but with less elaborated methods.

-The costs of PE screening in Table 2 require more details. Currently only a sum is represented. The cost prices for each separate test should be added (see row 1).

-The cohort for which the calculations were made is insufficiently described. The cohort is first mentioned in the results section. The wording “per cohort” is confusing. Were calculations carried out once or repeatedly for several cohorts?

-Please provide a rationale for carrying out one-way sensitivity analysis only. Additionally, why were all input parameters varied by the same percentage. Was this due to lack of data that would have allowed more specific assumptions for different input parameters?

-In the discussion the authors recommend including PE screening in reimbursed care in Germany. Whether PE screening should be reimbursed will depend on WTP for PE cases or stillbirths averted. This is a political decision. To justify this statement, I recommend presenting cost-effectiveness acceptability curves.

-The discussion would benefit from considering the role of the new screening algorithm in comparison to potential other alternatives, in particular novel maternal serum biomarkers (see: https://www.sciencedirect.com/science/article/pii/S2210778918307670)?

Minor points:

-Line 51: HELLP-syndrome: please spell out the first time this abbreviation is used

-Line 60: please specify which guidelines you refer to

-Line 90: please add a literature reference for FMF

-Line 169: it should read “one-way sensitivity analysis”

-Line 224: please specific what is meant by “at the second level”

-Line 260: please explain, why the percentage of women who develop preterm PE would change so significantly within 3 years

6. PLOS authors have the option to publish the peer review history of their article (what does this mean?). If published, this will include your full peer review and any attached files.

Reviewer #1: No

---

## [Author Response · Author response to Decision Letter 0]

8 Mar 2022

Response to reviewer´s comments

PONE-D-21-23946

Cost-effectiveness analysis of implementing screening on preterm pre-eclampsia at first trimester of pregnancy in Germany and Switzerland

PLOS ONE

We went through all author guidelines including those for file naming and adapted the manuscript´s style and file naming accordingly. 

2. In your Data Availability statement, you have not specified where the minimal data set underlying the results described in your manuscript can be found. Upon re-submitting your revised manuscript, please upload your study’s minimal underlying data set as either Supporting Information files or to a stable, public repository and include the relevant URLs, DOIs, or accession numbers within your revised cover letter. 

We have now included supporting information to our manuscript. It lists the input data necessary to repeat our analyses in tables, includes all probabilities, resource use, costs, ranges used for the one-way sensitivity analysis, and the distributions and standard errors used in the probabilistic sensitivity analysis. Also included in the supporting information are the data points used to build the figures showing the results of the one-way sensitivity analyses and the cost-effectiveness acceptability curves. 

3. We note that you have indicated that data from this study are available upon request. 

a) If there are ethical or legal restrictions on sharing a de-identified data set, please explain them in detail (e.g., data contain potentially sensitive information, data are owned by a third-party organization, etc.) and who has imposed them (e.g., an ethics committee). 

b) If there are no restrictions, please upload the minimal anonymized data set necessary to replicate your study findings as either Supporting Information files or to a stable, public repository and provide us with the relevant URLs, DOIs, or accession numbers. 

We now included the supporting information within our manuscript in the supporting information file (see above). 

4. PLOS requires an ORCID iD for the corresponding author in Editorial Manager on papers submitted after December 6th, 2016. Please ensure that you have an ORCID iD and that it is validated in Editorial Manager. To do this, go to ‘Update my Information’ (in the upper left-hand corner of the main menu), and click on the Fetch/Validate link next to the ORCID field. This will take you to the ORCID site and allow you to create a new iD or authenticate a pre-existing iD in Editorial Manager. Please see the following video for instructions on linking an ORCID iD to your Editorial Manager account: https://www.youtube.com/watch?v=_xcclfuvtxQ.

We included the ORCID-ID of the corresponding author in the Editorial Manager. 

Additional Editor Comments:

Please revise your manuscript according to the reviewer's advise and suggestions. Table 3 and 4 may be combined with one Table.

Thank you for the suggestions. Table 3 and 4 are now merged into one table i.e. table 3. 

 

Reviewers' comments:

Reviewer #1: This article describes a model-based cost-effectiveness analysis of screening for pre-term eclampsia compared to usual care in Germany and Switzerland. The question addressed is interesting, as the screening procedure is currently not covered by statutory health insurers in Germany and Switzerland and published economic evaluation studies for these countries are lacking.

Overall, the article is well-structured. However, the following points require further specification:

-Some background on the different health care and policy contexts in Germany and Switzerland (e.g., what does usual care include, who conducts screening etc.) would be helpful in the introduction. In line 95 an explanation should be added why PAPP-A was not recommended for Switzerland.

Thanks for pointing out the need for adding more context on PE screening in Germany and Switzerland. We added additional information on who conducts the screening and what the routine screening in usual care consists of. In the introduction, starting in line 62, it now reads: 

“In both Germany and Switzerland, antenatal care for pregnant women includes regular visits at the gynaecologist or, for pregnancies without complications, with a midwife. In the current standard of care, women at high risk of PE are identified through a routine screening during one of the first antenatal consultations, by means of analysing maternal characteristics and risk factors, as well as previous pregnancies and medical records.” 

The decision tree does not distinguish all possible scenarios. For example, false positive and false negative screening tests and aspirin vs. no aspirin among patients with a standard check-up are not included. Possibly, only summary data are available which do not allow more level of detail in the model. Please provide a rationale for the model structure.

The reviewer is indeed correct in pointing out that false positives and false negatives, as well as aspirin vs. no aspirin prophylaxis in routine screening, were not included in the decision-tree, due to limitations in the available data. It was therefore not possible to include more detailed information.

To provide the reviewer with a bit more background: With the current PE screening, the detection rate of preterm PE is almost 80%, at a false positive rate of almost 10% (Rolnik, 2017, Ultrasound Obstet Gynecol). Thus, also with the innovative PE screening, not all women at high risk are detected (false negatives) and some women develop preterm PE but were not detected as being at high risk (false negatives). In the analysis, this is included in the number of women developing preterm PE when being classified as at high risk or not. The false positives and false negatives are thus not included as separate branches in the decision-tree but ist included in the results. 

The same is true for the women in routine screening who do or do not receive aspirin prophylaxis when being at high risk. Currently, it is not known how accurately women at high risk of preterm PE are identified in routine screening, nor how many of them subsequently are being prescribed aspirin prophylaxis. For the cost calculations in the model, we used expert opinion that 80% of the women identified as at high risk receive aspirin prophylaxis. For the number of women developing preterm PE in routine screening we used the total number of women developing preterm PE as summary data for routine screening. In the manuscript we have explained this starting in line 116: 

“In PE screening, women were classified as at high risk for preterm PE or not. Women at low risk of preterm PE received standard antenatal care. Women at high risk of preterm PE were prescribed low-dose aspirin prophylaxis. All women in PE screening could develop preterm PE or not, and have a stillbirth, a live preterm birth (<37 weeks of gestation), or a live term birth. 

In routine screening, women could be classified as at high-risk based on maternal characteristics and might be prescribed low-dose aspirin prophylaxis. As no data was available on the probability of being classified as at high risk for preterm PE in routine screening and on subsequently receiving aspirin prophylaxis, we used summary data on the average number of women experiencing the outcomes stillbirth, live preterm birth, and live term birth. This was done irrespectively of a previous classification of being at high risk for preterm PE, as the available data did not allow for more level of detail. 

The impact of false positive and false negative screening results for both screening alternatives were included in the outcomes of both interventions, i.e., some women received aspirin prophylaxis unnecessarily, and some did not receive aspirin prophylaxis whereas they should. This results in a higher chance in both screening alternatives of developing preterm PE than when the prophylaxis had been received.”

The role of the variable “pre-term birth” is unclear. In the decision tree it seems that pre-term birth is an outcome, however it is not included in table 3 and 4. Pre-term births and still births averted seem a more meaningful outcome from the patient perspective than PE cases averted. Please describe the rationale for choosing PE cases averted as primary outcome.

Indeed, preterm birth, still births, and PE cases are the three outcomes mentioned in the decision-tree. We agree that preterm birth also should be reported in the tables 3 and we therefore included it in the new Table 3 (tables 3 and 4 combined). 

The reviewer is current in pointing out that preterm birth and still birth are very meaningful outcomes from the patient perspective. We feel that preterm PE by itself meaningful as well, as the diagnosis is very stressful for the mother and the families. In addition, it may lead to hospitalisation, early delivery (even though it might not be preterm), and a potential increase in risk of long-term outcomes. Thus, we find it difficult to decide which out these outcomes is most important and to decide how to weight quantity and severity. We have adapted the description of the outcome measures, which now read in line 132:

“The primary outcome measures were the incremental health care costs per women screened in PE screening versus routine screening and the incremental costs per PE case avoided. Secondary outcomes were the incremental costs per preterm birth avoided, and per still birth prevented.”

For the PSA we chose the ICER of costs/preterm PE cases avoided for the probabilistic analysis, as the preterm PE cases also include the increased risk associated with preterm PE of preterm birth and still birth. This now reads in line 202 onwards: 

“To analyse the uncertainty surrounding the results, a probabilistic sensitivity analysis (PSA) was conducted on the incremental costs per PE case avoided. This outcome measure was chosen since the main aim of the screening is to prevent cases of preterm PE, and as developing preterm PE also includes the higher chance of developing preterm birth or still birth, as well as the negative health consequences of preterm PE itself.”

In Figure 2 a legend would be helpful. I assume the (+) means that the path starting with standard check-up needs to be added here. 

Thank you for pointing out this omission, we added a figure legend. 

Information is missing about key assumptions regarding the relationship between events and costs. For example, one would assume that the rate of Caesarian sections is higher among patients with PE. 

When re-reading our manuscript with this information we indeed see this information is missing, thank you for pointing this out. From line 154 onwards we now explain how the outcomes are linked to the screening alternatives: 

“In both screening alternatives, women could develop preterm PE or not, depending on the screening accuracy to detect preterm PE. Without preterm PE, the probabilities of caesarean section, stillbirth, and preterm birth were identical to the averages in the respective countries (see table 1). When developing preterm PE, these were higher. For Germany, the probability of delivery by caesarean section with preterm PE was 99.2% [14], of stillbirth 0.74% [14], and of preterm birth 75.0% [14]. With preterm PE, the length of stay increased by 3 days for the mother and 16 days for the neonate [15]. For Switzerland, the probability of delivery by caesarean section of women with preterm PE was 61.9% (calculation based on [16]), of stillbirth 0.74% [4], and of preterm birth 75.00 [3]. For Switzerland, the extended length of stay for preterm PE patients was included using a DRG-code for a hospital stay with preterm PE.”

The rates of delivery with Caesarean section indeed were higher when developing preterm PE, for Germany, e.g., this is around 30% without preterm PE and 99% with preterm PE. We have restructured table 1 to make it easier for the reader to identify the probabilities per outcome based on having preterm PE or not. This should make it much easier to compare the percentages and to understand how the screening alternatives influence the number of patients developing preterm PE and how this influences outcomes. 

The term “interventions” is confusing. It would be clearer to speak about alternatives or more specifically about routine screening and innovative screening. According to my understanding screening does already take place in routine care, but with less elaborated methods. 

We agree that the term “interventions” was confusing. We have removed it throughout the manuscript and now use the term “screening alternatives”. What was previously “standard of care” is now termed “routine screening”, as it is correct that also in standard of care a screening takes place (based on maternal characteristics). 

The costs of PE screening in Table 2 require more details. Currently only a sum is represented. The cost prices for each separate test should be added (see row 1). 

To provide a better impression of the costs of PE screening we now included the costs of the biomarkers in table 2. For Germany, we used the ebm code 32362 for PlGF and PAPP-A of €19.40. Next to these, costs are incurred for drawing the blood sample, for the Doppler ultrasound, MAP, interpretation of the screening results, and providing advice to the patient. It is currently unknown how high the reimbursement for the screening would be, it was estimated around €60. In private practice, the prices charged for PE screening currently are around €130, but these are not regulated and thus differ among practices. Reimbursement by public health insurance is expected to be lower than the privately charged amount. Based on the estimation of the experts we talked to, it was estimated that 90€ is a realistic reimbursement to expect. As stated in the manuscript, it is not certain what would be the final reimbursement. 

For Switzerland, Sfr. 80 are for the PlGF measurement, based on the tariff code 1474.10, with the remaining parts of the screening being covered with Sfr. 70. We have updated the description of the PE screening costs in Table 2. 

The cohort for which the calculations were made is insufficiently described. The cohort is first mentioned in the results section. The wording “per cohort” is confusing. Were calculations carried out once or repeatedly for several cohorts? 

The reviewer is right that the term “per cohort” was confusing. We calculated all results as averages per patient, as well as per year for all pregnant women in the respective country, to enable decision-makers to identify how many cases of preterm PE, stillbirths, and preterm births can be prevented in their country on a yearly basis. From line 190 onwards we added explanation to the analysis section under methods:

“The results were presented as averages per woman screened and on the population level for all pregnant women per year in the respective country, which were 763,732 in Germany [17] and 84,759 in Switzerland [20, 21].”

Also, in table 3 we changed the wording from “per cohort” to “Results for all pregnant women per year”. 

Please provide a rationale for carrying out one-way sensitivity analysis only. Additionally, why were all input parameters varied by the same percentage. Was this due to lack of data that would have allowed more specific assumptions for different input parameters?

We looked again at the data used for the one-way sensitivity analysis. Where available, we now used a reasonable range when this was different from the +/-25%. Please see the S1 Table for the data and ranges used. 

We also added a probabilistic sensitivity analysis to our analysis. It is described in the methods section under “sensitivity and scenario analyses” starting in row 197. In the results we present the cost-effectiveness planes (figs 4 and 5) and cost-effectiveness acceptability curves (figs 6 and 7). 

In the discussion the authors recommend including PE screening in reimbursed care in Germany. Whether PE screening should be reimbursed will depend on WTP for PE cases or stillbirths averted. This is a political decision. To justify this statement, I recommend presenting cost-effectiveness acceptability curves.

We fully agree that based on the results for Germany, it is a political decision to decide on the willingness-to-pay for the prevention of preterm PE cases. To provide the relevant data for this decision, we now present CEACs for Germany and Switzerland in the results section. 

The discussion would benefit from considering the role of the new screening algorithm in comparison to potential other alternatives, in particular novel maternal serum biomarkers (see: https://www.sciencedirect.com/science/article/pii/S2210778918307670)?

We feel to discuss the role of novel serum biomarkers and the suggested article is a valuable addition to the discussion and have added a paragraph on this topic, starting in row 365:

In the prevention and management of preeclampsia after the first trimester, screening alternatives are available next to routine and innovative PE screening in the first trimester. For short-term prediction of preeclampsia, novel maternal serum biomarkers, such as the ratio between anti-angiogenic soluble fms-like tyrosine kinase-1 (sFlt-1) and PlGF taken between 18+0 and 36+6 weeks of gestation were found suitable to identify pregnant women with PE symptoms who require hospitalisation and women who can safely be monitored as outpatients [35]. A review by Schlembach et al. (2019) identified that angiogenic biomarkers to predict preeclampsia are cost-saving, among others by preventing unnecessary hospitalisation [36]. Thus, while PE screening is suitable for identifying women at high risk of preterm PE in the first trimester who benefit from aspirin prophylaxis, angiogenic biomarkers form cost-saving options for predicting the short-term risk of PE between 18+0 to 36+6 weeks of gestation.”

Minor points:

-Line 51: HELLP-syndrome: please spell out the first time this abbreviation is used 

Thanks for noticing us, this is done. 

Line 60: please specify which guidelines you refer to 

This was added and referenced. 

Line 90: please add a literature reference for FMF 

The reference was added. 

Line 169: it should read “one-way sensitivity analysis” 

Thank you, we have corrected the spelling. 

Line 224: please specific what is meant by “at the second level” 

We have changed to wording to make it easier to understand. With “second level” we meant in a second step, i.e. first conducting PE screening without UAPI, and then offering it to the 30% for whom the highest risk was found. It now reads: “Introducing contingent PE screening with offering the UAPI measurement to only the 30% of patients being at highest risk”.

Line 260: please explain, why the percentage of women who develop preterm PE would change so significantly within 3 years.

We expect that with time, prevention of preterm PE through aspirin prophylaxis would receive more attention, also as the innovative PE screening becomes available, and therefore, more gynaecologists would prescribe (or start to prescribe) aspirin prophylaxis to women with a high risk as identified in routine screening. However, this would indeed take a while. We have changed the description of this sentence in line 310: 

“The current study has a couple of limitations. First, the input data for the percentage of women who develop preterm PE in standard of care was relatively old [12]. It is possible that with time gynaecologists in routine screening would prescribe aspirin prophylaxis to an increasing percentage of women at high risk of preterm PE. Therefore, in the data for routine screening, fewer women identified as at high risk received aspirin prophylaxis than they would in current practice. We therefore adjusted the percentage by a relative 10% [13].”

---

## [Decision Letter · Decision Letter 1]

11 Apr 2022

PONE-D-21-23946R1Cost-effectiveness analysis of implementing screening on preterm pre-eclampsia at first trimester of pregnancy in Germany and SwitzerlandPLOS ONE

Dear Dr. Vrijhoef,

Thank you for submitting your manuscript to PLOS ONE. After careful consideration, we feel that it has merit but does not fully meet PLOS ONE’s publication criteria as it currently stands. Therefore, we invite you to submit a revised version of the manuscript that addresses the points raised during the review process.

We look forward to receiving your revised manuscript.

Kind regards,

Rashidul Alam Mahumud, MPH, MSc, PhD

Academic Editor

PLOS ONE

Journal Requirements:

Reviewers' comments:

Reviewer's Responses to Questions

**Comments to the Author**

1. If the authors have adequately addressed your comments raised in a previous round of review and you feel that this manuscript is now acceptable for publication, you may indicate that here to bypass the “Comments to the Author” section, enter your conflict of interest statement in the “Confidential to Editor” section, and submit your "Accept" recommendation.

Reviewer #1: All comments have been addressed

2. Is the manuscript technically sound, and do the data support the conclusions?

Reviewer #1: Yes

3. Has the statistical analysis been performed appropriately and rigorously? 

Reviewer #1: Yes

4. Have the authors made all data underlying the findings in their manuscript fully available?

Reviewer #1: Yes

5. Is the manuscript presented in an intelligible fashion and written in standard English?

Reviewer #1: Yes

6. Review Comments to the Author

Reviewer #1: The authors have addressed my questions and comments very rigorously. The main text, figures and supporting information are clear and easy to follow now.

One last minor point: pls clarify the [+] symbol in Figure 1 (decision tree). As mentioned in a previous comment, I assume that the symbol indicates that the branches following the standard check- up need to be added here. I am not entirely sure however, because of the different terminology: regular ckeck up vs. standard check up. There is still no legend included the figure.

7. PLOS authors have the option to publish the peer review history of their article (what does this mean?). If published, this will include your full peer review and any attached files.

Reviewer #1: **Yes: **Adrienne Alayli

---

## [Author Response · Author response to Decision Letter 1]

21 May 2022

Thank you for confirming that we have addressed your questions and comments. Regarding the last minor point and the legend for Figure 1; these were addressed in the revised manuscript, page 5, lines 114-115: 

[+]: The branches from above are repeated.

PE: preeclampsia, WGA: weeks of gestation.

---

## [Editor Report · Decision Letter 2]

13 Jun 2022

Cost-effectiveness analysis of implementing screening on preterm pre-eclampsia at first trimester of pregnancy in Germany and Switzerland

PONE-D-21-23946R2

Dear Dr. Vrijhoef,

We’re pleased to inform you that your manuscript has been judged scientifically suitable for publication and will be formally accepted for publication once it meets all outstanding technical requirements.

Kind regards,

Rashidul Alam Mahumud, MPH, MSc, PhD

Academic Editor

PLOS ONE
---

## [Editor Report · Acceptance letter]

20 Jun 2022

PONE-D-21-23946R2 

Cost-effectiveness analysis of implementing screening on preterm pre-eclampsia at first trimester of pregnancy in Germany and Switzerland 

Dear Dr. Vrijhoef:

I'm pleased to inform you that your manuscript has been deemed suitable for publication in PLOS ONE. Congratulations! Your manuscript is now with our production department. 

Kind regards, 

on behalf of

Dr. Rashidul Alam Mahumud 

Academic Editor

PLOS ONE